# 50 Years of Criticizing Religion: A Historical Overview of Norwegian Religious Education

Sebastian Tjelle Jarmer

MF Norwegian School of Theology, Religion and Society, 0363 Oslo, Norway; sebastian.t.jarmer@mf.no

**Abstract:** The critique of religion is hotly debated in contemporary media, legal and educational discourses. This review takes almost 50 years (1976–2022) of Norwegian research on the critique of religion in religious education as a point of departure to highlight how the discourse on the critique of religion is negotiated and represented. The review showcases the intimate connection between historical contexts and discursive repertoires through historical and critical discourse analysis. The analysis showcases that the discourse on the critique of religion is dynamic and diverse—starting mainly to appear through theological discourses referencing internal and hermeneutical critique before developing into more diverse discourses emanating from multiple actors and genres centered around critical thinking, source-evaluation, intercultural competence, and negative criticism of religion.

**Keywords:** critique of religion; religious education; Norway

## 1. Critique of Religion in Norwegian Religious Education; A Case Study

Critical thinking and critical inquiry have become important skills in school, work, and society writ large (Ferrer et al. 2019). But how should critical inquiry be extended towards religion? This article takes 50 years of research on Norwegian religious education as a point of departure to discuss the discourses on the criticism of religion. "Religious education" can refer to different matters depending on the geographical and historical context discussed. Here, it is used as a collective term referencing the historical and contemporary Norwegian school subjects that deal with the explicit teaching about religion, ethics, and worldviews in the 50-year period analyzed. The goal of this article is to contribute to the literature on religious education by systematizing an area of research that is underdeveloped (cf. Hammer and Schanke 2018, pp. 182–83; Löfstedt 2020, p. 5). This study is a summary of former research, but also builds the grounds for future research by highlighting diachronic developments in the field (cf. Krumsvik and Røkenes 2016, p. 62). An important part of the summary is to bring research that has been unsystematized, scattered, and buried in journals to the attention of scholars. The review can also be useful for teachers or teacher educators looking for concrete ways to handle the theme of critique of religion in the classroom (or contentious issues more broadly), as teachers seem to struggle with teaching about the critique of religion (Andersland and Aukland 2020; Löfstedt and Sjöborg 2020, pp. 143–45). Former research has emphasized the importance of being critical in religious education, as critique can showcase the multiplicity behind religious traditions and foster intercultural competence (Smith et al. 2018, p. 15; Jensen 2019, p. 36). Norwegian religious education is a diverse case study that has analytical value outside of the Norwegian context. The case illustrates how researchers from different ontological and epistemological frameworks try to rectify a pressing issue in contemporary society; how to criticize religion constructively. This is important because critique can effectively challenge power and change religious practices that perpetuate human suffering and oppression. It is also imperative to engage in nuanced criticism of religion, as unreasonable critique may reinforce stereotypical imageries. A balanced conversation about the role of religion is also

essential in post-secular contexts characterized by the increased visibility and contention around religion in both media discourses and the public sphere (cf. Löfstedt 2020, p. 2). Recent international research has also emphasized a new demand for scholarly discourses to engage in constructive criticisms of religion (see Franck and Stenmark 2019; Stenmark 2020; Lövheim and Stenmark 2020). The Swedish philosopher of religion, Michael Stenmark points to the importance of criticism of religion for liberal democracies:

> To have the right and possibility to criticize religions in public life is crucial for developing a healthy liberal democratic society . . . a liberal democratic society must allow people who adhere to different worldviews to criticize each other's religious or non-religious beliefs, values, and practices but still maintain relations of harmony and comity across diverse outlooks on human life and its ends within its boundaries (Stenmark 2022, p. 1).

Religious education can lay an important foundation in this regard by teaching pupils how to criticize religion in constructive ways.

Norwegian religious education is compulsory for all pupils in Norwegian primary schools. For pupils who choose a general study program, religious education is an obligatory subject in the third and last year of upper secondary school. The subject is geared both toward developing knowledge and shaping attitudes, and one of its main goals is to reduce the conflicts potentially produced by religion in a multicultural society (Toft 2019, p. 8). Until 1997, Norwegian religious education was confessional and had close ties to both Christianity and the Church. While formally becoming non-confessional in 1997, religious education was still considered by many to be "marinated in Lutheran Protestantism" (cf. Berglund 2013). Christianity's central place in the subject has sparked controversy and discussions (Lippe and Undheim 2017, pp. 11–15). Some argue that a confessional and empathic attitude towards religion has persisted after religious education was changed in 1997 (Anker and Lippe 2016, p. 261; Andreassen and Lewis 2017, pp. 1–6; Bøe 2020, p. 3). In this empathic framing, the critique of religion becomes problematic as it can be interpreted as undermining the development of respect and tolerance, which are seen as important goals for religious education. The critique of religion has nevertheless become an explicit part of the curriculum for religious education in Norway, being introduced in 2006. The criticism of religion might have been included in the curriculum due to the unprecedented attention that was paid to the destructive sides of religion after events such as 9/11, the Danish "cartoon crisis", and the numerous sexual abuses allegedly carried out by clergy of the Catholic church.

Teachers might face multiple challenges related to the teaching of critique of religion: How should teachers balance critique, respect, and tolerance? How should critique about religious adherents present in the classroom be taught? Can critique be constructive, or can it contribute to stereotypes and hate speech? Should teachers critique religious practices not in accord with the Norwegian school's value foundation, such as LGBTQ discrimination? This article shows how researchers of religious education have addressed these challenges by answering the following research question:

> How has the critique of religion been represented and negotiated in the Norwegian research on religious education from 1976 to 2022?

The analysis starts in 1976 as this is the year of the first influential publications (Asheim 1976) in the so called "pedagogics of religion", a tradition often taken as the starting point of Norwegian research on religious education (Lied 2006; Andreassen 2008a).

## 2. Methodology: Historical and Critical Discourse Analysis

This study incorporates elements from both historical and critical discourse analysis (cf. Fairclough 2010; von Stuckrad 2013). Certain tools from these methodologies are seen as constructive because they allow for an analysis of how actors in religious education research draw from each other, engage with other disciplines and instigate historical change through discursive negotiations. I analyze the development of the discourses on the critique

of religion in changing social, political, and historical contexts (cf. von Stuckrad 2013, p. 15). Concretely I look closer at three elements:

1.  The interplay between discourses and contexts, or what Michel Foucault calls the dispositive, which is understood as "the totality of the material, practical, social, cognitive, or normative 'infrastructure' in which a discourse develops" (von Stuckrad 2013, p. 15). I will pay particular attention to the curricular frameworks present in the 50 year period, as many of the texts analyzed here are written as a response to new syllabuses. I look closer at the curriculum introduced in 1974, *M74* (called *Mønsterplanen for grunnskolen*) as well as its revised edition, which was introduced in 1987, *M87*. Furthermore, I analyze the curriculum for 1997, *L97* (*Reform* 97). Lastly, I look closer at the newer curricula, first introduced in 2006 as *LK06* (*Læreplanværket*) and later revised and set in motion in 2020 as *LK20* (*Fagfornyelsen*). I henceforth trace the critique of religion through four rough time periods delimited by curricular change: 1976–1997, 1997–2006, 2006–2020, 2020–2022 (present).

2.  I also look at the interdiscursive dynamics in the texts by locating concrete tendencies in the discourse such as where the critique is positioned from (epistemologically/ontologically), and what sorts of criticisms (hermeneutical, interreligious, internal, external, see under) are fronted in the different periods. I also locate the pedagogical and didactical arguments underlying the researchers' call for criticism of religion in the classroom. When relevant, I identify how the discourses are interdiscursively recontextualized (cf. Fairclough 2010) from other discursive fields (such as media or politics) into religious education research, for instance when scientific discourses are introduced to critique religion from a biological standpoint in the classroom.

3.  Lastly, I analyze the (manifest) intertextual references, or the direct citations in the discourses to showcase how authors build on or challenge each other (cf. Jørgensen and Phillips 2002, p. 73).

The article analyses different *scientific power texts*, meaning texts which instruct, prescribe, and give advice about teaching the critique of religion (cf. Andreassen 2008a, pp. 49–50). Power texts can be labeled "push media" as they attempt to direct the reader in a specific direction through mono-cataloging discourses on religion (Andreassen 2014, pp. 1–5). Some power texts may have much power in a given time and space (such as widely used textbooks), while others do not have so much power (barely read article entries). However, the texts all have the *potential* to become powerful since they are directed toward the future (cf. Fairclough 2010).

I analyze the publications on the critique of religion as a "field": a social domain entailing semi-formalized routines (Chouliaraki and Fairclough 1999, pp. 101–8). Actors in a field, such as sports, politics, or for our sake (publications about) religious education, have a common habitus that shapes their goals and practical sense of "the game" (Chouliaraki and Fairclough 1999, p. 101). In our context, "the game" is about instructing the practices of the critique of religion from different perspectives. The actors in the same field struggle for dominance of discourses (Jørgensen and Phillips 2002, pp. 72–73). This involves attempts to stabilize the order of the discourse by privileging some representations and hiding others. Three elements relate the actors in the field analyzed here to each other: They discuss religious education, produce power texts which instruct teacher practices, and represent the critique of religion in different ways.

In historical discourse analysis, all texts in a field can seldom be analyzed. I have therefore made strategic choices about which texts to include, to shed light on central tendencies in the discourse present at the time (see Jørgensen and Phillips 2002, p. 147). I have manually searched leading journals, introductory books and followed references.

The critique of religion is considered a floating signifier, meaning it is an important "sign" in the "field" of religious education research that different actors try to define (Jørgensen and Phillips 2002, p. 28). Without retroactively defining what the critique of religion is, I have used certain theoretical insights as "sensitizing concepts" to guide analysis

regarding what critique of religion might be. This includes tentatively defining the critique of religion as a negative evaluation, or a representation that aims at changing something (see Lövheim and Stenmark 2020), a representation that favors one perspective and argues against another (Skirbekk 2011; Søvik 2018), or a reflexive endeavor entailing analysis and scrutiny of a demarcated subject. Critique of religion is also often conceptualized as either internal/external/interreligious depending on the positionality of the criticizer and the criticized (Andreassen 2016, pp. 138–42; Stenmark 2020, pp. 18–21).

### 3. Internal, External, and Hermeneutical Critique: Theologically Informed Religious Education from 1976–1997

Before engaging in the analysis of the discourse on the critique of religion between 1976–1997, we need important contextualization of Norwegian religious education and the presence (or lack thereof) of the critique of religion in the official documents of the time period. Until the 1800s, the Norwegian school was Christian and ended in confirmation (Lippe and Undheim 2017, pp. 12–13). Facing globalization and increased diversity, alternative subjects were opened to non-Christian students in both 1974 ("Worldview-orientation", M74) and 1987 ("Other religion- and worldview education", M87). M74 had some potential references to the critique of religion, as it is specified that pupils in sixth grade should discuss "current problems". M87 also mentions themes that might cover the critique of religion, mainly through discussions of "evil and suffering in the world", "conflict resolution" (4–6 class), and "problems with the evil in the world" (7–9 class) (see Skrunes 1999, pp. 102–4). Religious education remained confessional until 1997. The Christian/confessional religious education model is the context for religious education from 1976–1997. The analysis starts in 1976, as it is the year of the first textbook in the very influential "tradition" of religious education research, later called the "pedagogics of religion". The "founders" of the pedagogics of religion in the Norwegian context, Ivar Asheim (1927–2020), and Sverre Dag Mogstad (1947–), both have ties to MF school of theology, a private school of theology established in 1907. They built their instructions and theorizations on pedagogical theory grounded in Christian-theological reflections. Through individual and collective publications and frequent cross-referencing (Asheim and Mogstad 1987, pp. 55, 134; Mogstad 1990, p. 71) they developed an intertextual "web" that reinforced and legitimized their perspectives in the "field" of religious education research (cf. Fairclough 2010). Asheim and Mogstad's success can clearly be shown in their persistent utilization in education, as many schools used their textbooks as curricula until the 2000s (Lied 2006, p. 187; Andreassen 2014, p. 6). The books analyzed here are textbooks written for an audience of both researchers and teachers.

This discourse analysis showcases that mainly three discourses on the criticism of religion are present in the research scrutinized between 1976–1997: Internal, external, and hermeneutical critique. *Internal criticism*, that is criticism from the same religion that is criticized, is concerned with a negotiation about what Christianity *is* and *should be represented as* in religious education. *External criticism*, that is criticism from outside of the religious tradition that is criticized, is evoked, and engaged with through interdiscursive references to social systems such as media and politics, as well as critique leveled from Christianity's "enemies". Lastly, the discourse in the period includes a *hermeneutical critique* of religion, which can be defined as a critique of religion that highlights a multiplicity of interpretations that challenge predisposed understandings of religion.

Overall, Asheim and Mogstad discuss the critique of religion to a very limited degree. Especially lacking are discourses that highlight the negative role that religion can play in society. This reflects the interplay between discourse and dispositive, as the theologically rooted and confessional nature of the religious education-subject at the time, as well as the lack of mention of the criticism of religion in the curriculum, may have disincentivized certain critical discourses from being fronted. However, Asheim and Mogstad do suggest that teachers should help students develop an adequate understanding of their own religious beliefs through *internal criticism of Christianity*. Teachers are supposed to guide pupils by

developing their ("wrong" or "flawed") religious mindset and socialize pupils into stages of "religious development" (Asheim and Mogstad 1987, pp. 122–23; Mogstad 1990, pp. 63–70). Teachers must "help" pupils on the right path by differentiating between different religious dimensions, such as faith, ethics, and history. A cross-disciplinary "problem-centered" didactic revolving around existential themes is also suggested to accompany the cognitively focused bible studies (Asheim 1976, pp. 93–96; Mogstad 1990, pp. 99–104). By putting existential reflection on the agenda, Christianity is instrumentalized and actualized. Two discourses on the critique of religion are thus opened: A critique of "the old traditions" that are no longer relevant; and a critical discourse of the value and relevancy of religion in contemporary society. Such discourses might result in what Simmel (e.g., Simmel 1976, p. 259) called "post-Christianity", where discourses are fixed on instrumental and not historical grounds.

In the pedagogics of religion, the critique of religion is also related to *hermeneutical* and *external* criticism through engagement with different interpretations of Christianity. Pupils are instructed to not only approach texts that they already have a sympathetic relationship with, as this can result in "blindness" to the text's flaws. Henceforth, pupils should also engage with texts from their "enemies"—that is, a critique that emanates from external sources. Our enemies often have striking criticism of our beliefs. Good criticism requires great understanding, according to Asheim. This should not be underplayed, because "there has been written a lot of pertinent [criticism] about the church by its enemies" (Asheim 1976, p. 109). It seems as if Asheim advocates for a dialectical reading of Christianity's texts, both engaging in discourses emphasizing a text's strengths and flaws. The critique of religion thereby becomes a synthesizing practice embedded in nuanced hermeneutical understandings.

Asheim and Mogstad also engage with external critical discourses that try to address the authoritarian and oppressive crosscurrents in religion through education. These external critiques are directed towards "religion" broadly and not necessarily Christianity specifically. Through discussions of "critical pedagogy" (a pedagogical framework from the 1920s with roots in the Frankfurt school), Asheim and Mogstad discuss how education could contribute to developing students' critical view of contemporary society. They also draw on Marxist-Leninist perspectives, which are more concerned with challenging class structures through education. Lastly, through dialogue with German pedagogics of religion, they reflect on the emancipatory dimensions of education (Asheim and Mogstad 1987, pp. 13–14, 21, 163). The goal of these discourses is to critique ideology and reveal how education is influenced by the school's structures as well as socioeconomic factors. Importance is also given to the liberation of the human mind from predefined authorities, and continuous struggles to change the status quo. The authors do not identify themselves with critical pedagogy and are skeptical of its central tenets (ibid., pp. 21–22, 163). They nevertheless give a critique of religion credence through "critical perspectives" by representing it as relevant in the discussion of religious education and highlighting critical pedagogy as important in the grand pedagogical tradition.

Writing at the same time as Asheim and Mogstad, the theologian and sociologist Ole Gunnar Winsnes (1940–2021) became a central actor in the tradition of religious education research later called the "Winsnes"-tradition. This influential tradition is still considered under the umbrella term "pedagogy of religion", but retains a *pedagogical* and not theological focus. The aim was to create an alternative religious education based on contextual and empirical consideration (Lied 2006, p. 168). Winsnes has been an influential voice in the "field" of religious education, inspiring a wide range of empirical research (Lied 2006, p. 175).

Although operating under the same sociocultural and political circumstances as Asheim and Mogstad, Winsnes is considerably more skeptical of the status quo. Winsnes writes extensively about the apparent crisis that religious education is in at the time. Students find teaching to be childish, dogmatic, authoritarian, non-relevant, and archaic. Conversely, they want a religious education that addresses existential themes, current

events, and even confrontation (Winsnes 1984, p. 55). The pupils demand a more analytical and critical education. These tendencies must be taken seriously, according to Winsnes.

To create a more analytical religious education, Winsnes wants to engage with external criticism of religion from other social systems through a truth-seeking *community of inquiry*. Here, both teachers and pupils approach a text through continuous listening and questioning (Winsnes 1984, p. 41). In such a framework, neither teacher nor pupil is "above" the text but engages with it through her own experiences. Such readings can and *must* be critical, as Winsnes is fully aware that religious education no longer can rest on the assumption that certain religious beliefs or statements are true—a point which he reiterates again and again. Religious education must therefore make Christianity trustworthy, as it "is important that [religious] knowledge is supported by social structures, for instance generally accepted opinions, attitudes and behaviors" (Winsnes 1984, p. 27). Religious education must henceforth engage with the critique that is leveled against religion through discourses from other social systems, such as law, media, and economics. Christianity must be presented as a "thinking belief". This means that pupils must be able to intellectually make sense of the Christian faith and inhabit a critical and reflexive stance toward the Bible (Winsnes 1984, pp. 61, 66). Religious education can thereby help to legitimize religion in contemporary Norwegian society through bracketing predisposed judgments about the authenticity of certain practices and beliefs and by engaging with external discourses critical of Christianity.

Winsnes also discusses the critique of religion through textual and hermeneutical criticism. Text- and bible studies have historically challenged "official" versions of Christian doctrines. Winsnes wants religious education to engage with such criticism through historic and linguistic analysis. Pupils must try to understand the author's "horizon of understanding", while simultaneously relating the text to their current situation. But in this process, they are forced to acknowledge the importance of interpretation, which leads to a series of new questions: "What is true in the Bible?" and "What is Christianity?". Pupils are consequently confronted with the internal problems and contradictions in the bible, for instance regarding the uncertainty around the resurrection of Jesus (Winsnes 1984, pp. 133–34). A critical engagement with textual hermeneutics thus exposes competing discourses on Christianity, while also setting the stage for negotiations of what religion *is*.

### 3.1. Critique of Religion in Integrative Religious Education: 1997–2006

After being confessional and centered on Christianity for decades, religious education was completely changed in 1997. The old model was replaced by a non-confessional *integrative* religious education, where students with different backgrounds come together to learn about religions and worldviews (Alberts 2012). The new subject, *Christianity with an orientation about religion and worldviews* (KRL) was described as an "expanded subject of Christianity" which also emphasizes other "living religions" such as Judaism, Islam, Hinduism, and Buddhism (Andreassen 2016, p. 68). The critique of religion was arguably present in the curriculum through references to freedom of religion and historical mentions of Marx, Sartre, and Freud (cf. Andreassen 2016, p. 137).

I look closer at three different authors writing between 1997–2006, which reflect different positions in the discourse. My sample is informed by Andreassen's (2008a) thorough analysis of textbooks written after 1997. Two of the discourses discussed here are exemplified by the textbooks of Skrunes (1999) and Stabell-Kulø (2005), which according to Andreassen can be considered "centrifugal" voices in the discourse on religious education. This means that they challenge the dominant discourse that permeates institutions and official websites at the time of their writing (Andreassen 2008a, pp. 57–58). While many texts could be chosen to reflect this dominant discourse, I analyze the textbook edited by Sødal (2001) here,[1] because it addresses the critique of religion in explicit terms. In contrast to Andreassen (2008a) who postulates that an empathic and uncritical discourse dominated in the period after 1997, I find a new diversity of discourses on the critique of religion, especially from a diachronic perspective. The critique of religion is related

to a wider range of other discourses such as *perspectivity* (inside/outside distinctions), source evaluation, and critical thinking. Through recontextualizations, reductionistic "scientific" explanations of religion are introduced into the classroom. A new form of striking criticism also appears after 1997, intimately connected with the new integrative religious education where pupils get to know each other's perspectives. This criticism, which we can call integrative/interreligious criticism, appears through competing religious narratives about the same events. Further, some discourses double down on apologetics to *defend* religion from external criticism. Lastly, some attention is given to a *negative* critique of religion which highlights the need for religious education to actively criticize oppressive religious practices.

The textbook written by the professor (and former principal) at the private Christian school *NLA*, Njål Skrunes can be considered a centrifugal publication in the discourse because it is based on a theocentric (and Christian) pedagogy reflecting "Gods deeds in creation and redemption" (Skrunes 1999, pp. 53, 59). This might be considered a problematic foundation for non-confessional religious education, which might be why Skrunes is not a central figure in the discourse on religious education research. Skrunes also had limited influence on teachers' practices as he is seldom included in curricula for higher education (cf. Andreassen 2008a). It is nevertheless fruitful to discuss his textbook here, as it illustrates the persistent negligence of a critique of religion that highlights the negative aspects of religion in religious education. Skrunes also exemplifies a negotiation of what Christianity *should be* in religious education through internal criticism of religion, but interestingly bases his critique on conservative readings of the bible. Drawing from his theological background, Skrunes wants teachers to engage in *apologetics* and defend Christianity against the criticism of religion.

Building on the theological discourses that came before him, Skrunes frequently draws from the publications of Asheim and Mogstad. Like his theologian peers, Skrunes pays very little attention to the critique of religion. Skrunes does acknowledge that "the syllabus states that students should be able to critically evaluate both the subject matter and their own and others' points of view" (Skrunes 1999, p. 141), but this is not discussed in any meaningful detail. Conversely, it seems as if Skrunes is generally pessimistic regarding critical inquiry toward Christianity. In fact, he says that the teachers must defend Christianity as "spiritual values" appear to be in a "cultural headwind" (Skrunes 1999, pp. 173–74).

Concretely, Skrunes urges teachers to be ready to answer external criticism about the "Christian miracles". Such critique of religion will plausibly come from media discourses, the home, friends of pupils, and science (Skrunes 1999, pp. 186–89). Teachers must be aware of criticism from the natural sciences, which tries to explain the Christian wonders through "cause-and-effect". "Scientific objections" must not be ignored, as they can easily "cause doubt" in pupils (Skrunes 1999, p. 187). Therefore, the pupils must get help with "intellectual explanations" of religion (echoing Winsnes). However, teachers must always remember that "relationships with Jesus can never be based on 'external demonstration'", but are to be based on "belief" and "trust" in God (Skrunes 1999, p. 188). Skrunes therefore deems it necessary for teachers to be aware of various external criticisms of religion, but ultimately deems such critique irrelevant as the truth claims of the Christian religion cannot be repudiated by "external demonstration". This is a discursive strategy to protect Christianity from inferences that may "cause doubt" in pupils by making external criticisms inconsequential.

Skrunes also indulges in his *own internal critique* of religion, concerning what Christianity in school should be represented as. He disagrees with the notion that religious education should be "instrumentalized" in the sense that moral dilemmas "relevant at this time" are taken as the point of departure for discussions. Such twisting of religion makes Christianity obsolete, according to Skrunes. Instead, the classroom should engage in discussions of "controversial questions of morality" with diverging explanations (Skrunes 1999, p. 132). Skrunes is here in direct opposition to former dominant discourses in the field of religious education research perpetuated by his theologian peers, which illustrates

that former hegemonic discourses are not unequivocally orderly and can be negotiated. In these discursive struggles, a critique of what true Christianity should be depicted as is delineated.

Like Skrunes, the scholar of religion Stabell-Kulø's (2005) textbook had limited influence after the curriculum changed in 1997. The book builds mostly on religious studies perspectives, and raises numerous (purportedly) "new" questions regarding themes such as the essentialization of religion, multireligious societies, and secularization. In contrast to Skrunes, Stabell-Kulø's (2005) textbook is adamant about the role of the critique of religion in religious education. Through what we can call *negative critique*, Stabell-Kulø frequently emphasizes the negative impulses that religion can play in society. Such criticism has been neglected in the field of religious education research discussed so far. According to Stabell-Kulø, teachers of religious education must engage in unambiguous criticism of oppressive behavior influenced by religions. He upholds that we must not "underestimate that religions always have lit their bonfires for heretics" and that people "with religious eagerness has tortured and slaughtered both disbelievers and apostates" (Stabell-Kulø 2005, p. 30). Conclusively, religious education should not encompass a "naïve" conflict-reducing ideology (Stabell-Kulø 2005, pp. 18–19). This might be an interdiscursive critique of the authors in the pedagogics of religion who often view religion in empathic terms, without intertextually negotiating with them through stated references.

Stabell-Kulø also discusses various external critiques of religion. These criticisms we can label reductionistic or "science-based" discourses on religion, which might be interpreted as critical and challenging for religious adherents. In contrast to Skrunes, who warns that teachers should be aware of these scientific discourses, Stabell-Kulø urges teachers to introduce them. One such discourse could be exemplified in "memes", defined as "ideas [ . . . ] that are spread as a sort of self-replicating virus from one brain to the next, often with explosive speed" (Stabell-Kulø 2005, p. 33). It is important not to neglect memes as an important explanation for religion just because they can "rock" with religious and philosophical ideas, Stabell-Kulø posits. Scientific discourses are thus "recontextualized" (Fairclough 2010, p. 11) from their former fields (science) into the field of religious education research, in the process suggesting certain critical "explanations" for religions not perpetuated by religious adherents themselves.

The last textbook discussed here is edited by scholar of Christianity Sødal (2001). It was according to Andreassen widely utilized in schools and referenced in online discourses and therefore inhabited a dominant position in the discourse on religious education research at the time. In this dominant discourse, little space has been given to critical perspectives according to Andreassen: "A terminology for religious conflicts [ . . . ] has never been allowed to emerge, since professional thinking has been based on religion as unambiguously good" (Andreassen 2008a, p. 258). This is, however, not completely correct if we look more closely at Sødal's text. Critique of religion is for instance considered in positive terms in a passage about "indoctrination":

> The opposite of indoctrination will be critical reflection [ . . . ] this is an important objective for religious education. The pupils should not without criticism accept everything the various worldviews stand for. But the pupils' eventual critique should be based on factual knowledge about the tradition. Moreover, it is important that the different worldviews are presented equally—also when it comes to critiquing. If, for instance, Jehovas' Witnesses, Mormons, or Islam is criticized, the same type of critique should be leveled against Hinduism, Christianity, and Humanism. An education that gives room for critical reflection should work as a good vaccine against illegitimate indoctrination (Sødal 2001, p. 37).

Critique should in other words be based on "facts" and be presented "pluralistically", although it is not specified how a critique of specific traditions (i.e., caste in Hinduism) could also be leveled against other religions.

Critique of religion is also tied to what is termed "existential confusion" by Sødal and colleagues. Existential confusion about "what to believe in" might arise when pupils

are confronted with a diversity of religions and worldviews which can be interpreted as a "problematization of [some pupils'] own beliefs" (Sødal 2001, p. 141). This, we might call an *integrative* or an *interreligious critique of religion*, where different religious narratives and traditions dissent. Displaying each religion and worldview as equal arbiters of truth might result in questions like: "How can I know which God is the 'right' one?" (Sødal 2001, p. 141). Such "existential confusion" might be particularly challenging within religions that have direct contradictory accounts of the same events: Was it Isaac (Christian account) or Ismail (Islamic account) that was demanded sacrificed by Abraham? A more direct interreligious critique may also crop up in aesthetic formations with critical representations of other religions, for instance in Christian art where Jews are caricatured during Jesus' execution (cf. Sødal 2001, pp. 188, 260). We can therefore differentiate between two forms of interreligious critique: One implicit, where different religious narratives dissent without necessarily addressing each other (the Isaac/Ismail example), and one *explicit* interreligious critique where one religion openly criticizes the other (e.g., the caricatures of Jews in Christian art). Interestingly, the "existential confusion" that may be triggered by interreligious criticism in religious education is not necessarily deemed to be a problem, as "pupils become challenged to reflect on what they believe". Furthermore, they can also gain a "greater understanding about other people's beliefs [ . . . ] the plural reality will therefore not come as a shock later" (Sødal 2001, p. 141). Interreligious critique, or seeing a matter from another religious perspective, therefore develops intercultural competence.

To some degree, Sødal's book also engage in a *negative critique* of religion. The authors contend that schools cannot be "neutral" to violence or discrimination but must criticize and fight against racism, gender inequality, and genital mutilation done in the name of religion (Sødal 2001, pp. 31, 150). It is suggested that critique or judgment of religion can be posed from liberal ideals. The critical discourse is therefore normatively positioned *a priori* from a certain standpoint anchored in discursive repertoires (democracy and human rights).

Critical judgment is also related to awareness of genre. Pupils cannot be "critical in a constructive manner" if not the critique is differentiated towards the genre in question, the authors contend. They imply that we cannot judge a "mythological worldview" with the same approaches we do in STEM (Sødal 2001, p. 239). The argument seems to be that "religion" must be considered by other epistemological criteria than that of "science", a position which is not unusual in apologetic discourses (cf. Busk and Crone 2008, p. 9).

Sødal also relate the critique of religion to the academic "outsiders" and "insiders" views. From the academic "insiders" perspective, the teacher "tries to describe [religion] as accurately as possible seen from the inside" (Sødal 2001, p. 130). This perspective may co-structure critical voices from the tradition in question. Martin Luther can be represented as a brave (internal) critic of the catholic church "that dared to protest" in the name of liberation (Sødal 2001, p. 264). On the other hand, the academic "outsiders' perspective" describes religion as "correctly as possible without taking into consideration what a [religious] tradition teaches about the same events" (Sødal 2001, p. 130). For instance, a plausible hypothesis about Muhammed's revelation poses that he was influenced by his own experiences, such as stories from the Bible, Jewish legends, and other contemporary traditions. It is suggested that religious education should be differentiated based on age, becoming more and more critical as education progresses, as critical reflection is:

> Conditioned by the ability to see your own and other beliefs from an academic outside perspective. But this is not to be expected by the younger pupils [ . . . ] For the young pupils, it's important to be confident in their own identity, not to reflect critically on it [ . . . ] the outside perspective can be introduced gradually. In tenth grade, this will be the most important perspective (Sødal 2001, p. 131).

It is specified on multiple occasions that critical thinking and logic are important for pupils in later years of education (e.g., Sødal 2001, p. 115). However, it is not a truism that "young pupils" should not "critically reflect on their identity". This fixing of the discourse is consciously perpetuated by the textbook's authors to cement young

pupils' religious identities, perhaps reflecting some of the empathic framing discussed in Andreassen (2008a).

Sødal also relate the critique of religion to hermeneutics and the evaluation of sources. When analyzing a source, the pupils must look at the commercial, factual, confessional, and *critical* sides of the source to get a full picture of the discourse (Sødal 2001, pp. 229–30). This involves engaging with "the angry sides" of texts which are "often against or critical to religions or worldviews [ ... ] here one can find valuable information" (Sødal 2001, p. 230). This way the critique of religion is anchored in the polyphony of interpretations possible in the analysis of a given discourse, echoing the pedagogics of religion call to engage in hermeneutic critiques of religion through dialogue with "Christianity's enemies".

To sum up, new voices emerged in the field of religious education research after the introduction of the new subject in 1997. Former theological discourses are disrupted and renegotiated internally. New voices and themes emerge, related to themes such as critical thinking and source evaluation. Through recontextualizations, scientific discourses are introduced into the classroom. Integrative and negative critique is also introduced after 1997. This new diversity in the discourse is only a foreshadowing of the explosion of voices flourishing after the introduction of the new curriculum: LK06.

### 3.2. New Stratifications and Intertextual Battles; 2006–2020

In 2006, a new curriculum aimed at bettering the quality of the Norwegian school was introduced. In this process, religious education was heavily discussed, partly because Norway was convicted for breaching Article 2 in Protocol 1,[2] and partly because of the unclear position Christianity inhabited in religious education. Between 1997 and 2008, religious education was changed three times: in 2002, 2005, and 2008. For our sake, some changes are of particular interest. In 2002, the critique of religion was explicitly mentioned as pupils in the 10th grade should know "modern critiques of religion". It was elaborated in 2005 that pupils should "present examples of the critique of religion from different worldview traditions" (Andreassen 2016, p. 138). In upper secondary school, pupils had to "give an account of and evaluate criticism of various forms of religions and worldviews" and also "discuss cooperation and tensions between religions and worldviews, and reflect on the pluralistic society as an ethical and philosophical challenge" (Kunnskapsdepartementet 2022b, p. 3). With the explicit mention of the critique of religion, a wide diversity of scholarship discussing the theme emerged. New voices from various disciplinary backgrounds joined the debate, now also discussing the critique of religion through standalone articles and book chapters. To trace the interdiscursive negotiations in this period (2006–2020) I have sorted the publications into six analytical "sub-discourses". Each analytical category is exemplified by one or more authors. The sub-discourses display a wide variety of approaches to the criticism of religion, through different ontologies and epistemologies ranging from critical realism to hermeneutics. I will briefly summarize what differentiates each sub-discourse before moving on to a more in-depth analysis of each.

1.  The first discourse I call the *outside-perspective discourse*, which is mainly based on theories from religious studies and is exemplified by scholar of religion Bengt-Ove Andreassen. From the outside perspective, the teacher considers religion as a cultural phenomenon that should be analyzed as any other unit of analysis, just like "politics" and "history". Andreassen discusses the critique of religion in his introductory book to religious education (Andreassen 2016), but also writes about the theme in chapters of anthologies (Andreassen 2010) and journal entries (Andreassen 2008b, 2009). His works have been hugely influential in the Norwegian discourse of religious education. Andreassen is especially known for challenging the empathic framing of religion perpetuated in the pedagogy of religion (see over).

2.  I have gathered three authors in the second sub-discourse called the *critical discourse*. The authors here are philosopher Gunnar Skirbekk, who discusses the theme in articles, books and chapters (Skirbekk 2009, 2011, 2021), theologian and teacher

educator Øystein Brekke, who have published two articles on the critique of religion (Brekke 2018, 2020), and the systematic theologian Jan-Olav Henriksen who writes journal articles (in the journal *Religion og livssyn*, a "journal for teachers of religious education") on the subject (Henriksen 2005, 2012). The authors here are heavily influenced by Jürgen Habermas and Immanuel Kant. I have called the authors working in this discourse "critical", which references both critical theory (Habermas 2006) and Kant's overarching "critical project". In contrast to Andreassen, this sub-discourse largely *builds on* rather than challenges the theologically inspired pedagogics of religion.

3. The third sub-discourse emphasizes multicultural pedagogy and is therefore labeled the *multicultural discourse*, exemplified by scholar of religious education, Espen Schjetne, who writes about the theme in a chapter in an anthology (Schjetne 2014). Schjetne criticizes authors in the outside-perspective discourse and the critical discourse for having too simplistic discussions of the criticism of religion, and emphasizes the need to indulge more closely with power dynamics in the classroom. Schjetne's contribution is also the first that warns against the *dangers* of introducing unnuanced criticism into the classroom.

4. The fourth discourse I call the *critical-realist discourse*, which discusses the critique of religion with reference to critical realist ontology and epistemology, exemplified by two articles written by systematic theologian Søvik (2011, 2018). The starting point for a discussion about the critique of religion is here twofold. Firstly, pupils must assume that *something is real* (ontological realism). But they must also acknowledge that their knowledge of the world is influenced by their own perspectives (epistemological relativism). However, this does not mean that "anything goes"; some statements are relationally more accurate than others. Through discussions in the classroom, pupils can gauge which criticism is founded on sound premises and valid argumentation.

5. The fifth discourse I call the *praxis discourse* because the authors of the textbook analyzed here, university teacher and teacher educator Kari Repstad and coordinator for teacher education, Repstad and Tallaksen (2014) specify that they are mainly concerned with teachers' practices. The book is filled with suggestions about teacher practices inspired by concrete experiences from the authors. This also applies to the criticism of religion, which is discussed in some detail.

6. The last discourse I call the *hermeneutical discourse* exemplified by scholar of religion Kåre Fuglseth, who has written one article on the theme (Fuglseth 2018). Fuglseth largely bases his discussion on phenomenological perspectives and hermeneutics, especially emphasizing the grave importance of prior knowledge when criticizing religion.

Andreassen (2016, 2008b, 2009, 2010, 2017) has written extensively on the critique of religion in what I call the outside-perspective discourse. In his Ph.D. (Andreassen 2008a), he levels harsh critiques against a phenomenological and resource-oriented pedagogy that universalizes religion and considers it a common good. Instead, he wishes to orient religious education around an epistemology of religious studies based on a critical "outside perspective". In contrast to the former discourses, we can in Andreassen's formulations witness a destabilization of the formerly dominant "order of discourse" by *explicitly* breaking from the pedagogics of religion. This break is also clearly exemplified by Andreassen's insistence to focus on a negative critique of religion by highlighting the destructive sides of religion.

In Andreassen's trajectory, the critique of religion is important because it opens a space where different critical perspectives can be discussed. As such, religion does not only represent a positive force. Andreassen also wants pupils to have different "boxes" where they can place utterances as *certain* criticisms of religion, and therefore differentiates between external "classical normative" and internal "religious" critique of religion. In external critique, the discourse emanates from non-religious actors. Examples here are the so-called "masters of suspicion", such as Freud, Marx and Nietzsche, or modern critics, such as Dawkins, Dennett, and Harris. He posits that a challenge regarding teaching about

external critique is to teach about something anti-religious, without letting the teaching become anti-religious. Internal critique concerns the reinterpretation or rejection of religious ideas from individuals within a religious tradition. Andreassen gives Luther and Buddha as examples here. They are important because they highlight religious diversity and religious traditions interacting. Internal critique can also show diversity, according to Andreassen. He thinks that fundamentalism might also be considered an internal critique of religions, based on going back to the "sources".

Andreassen's clearest break from the pedagogics of religion may be exemplified by his advocacy of so-called conflict perspectives when teaching about the critique of religion. A conflict is a clash between two interests. It involves dissent and contrasting images about a common theme (Andreassen 2008b, pp. 10–11). A conflict might introduce a hermeneutical or interreligious critique of religion as they display competing interpretations of the same events. Conflict perspectives might also provide ample examples of "internal criticism of religion" through representations of adherents who fight over "right interpretations" of religious truths (Andreassen 2008b, pp. 10–11). Thus, conflict perspectives also show that religions have continuities and ruptures, similarities, and differences, as well as highlight which parts of religion are considered intrinsic to labels such as "Christianity" and "Islam". Conflict perspectives also show what role religions can play in contemporary problems, according to Andreassen (2010, p. 78). Drawing on a discussion of Klafki's reflections on *Bildung* and key problems (that is, pressing matters that must be addressed in a specific time and space, such as global warming, immigration, digitalization, etc.), he says that teaching must be more than a historical overview of older critics of religion. Teaching must dare to engage with current events in politics and science, which also will make the teaching more relevant for the pupils (Andreassen 2010). Lastly, what is also very important for Andreassen, is that conflict perspectives allow students to take a meta-theoretical standpoint regarding where a critique of religion is positioned from (Andreassen 2010, p. 77). Andreassen, in summary, advocates for a critique based on conflict perspectives, categorical descriptions such as outsider/insider critiques, and a meta-perspectival awareness.

The first researcher in the critical discourse is exemplified by philosopher Skirbekk (2009, 2011, 2021). Like the other authors in the critical sub-discourse, Skirbekk is heavily influenced by Habermas. From him, he insists on deliberative discussions and a modernization of religious consciousnesses in religious education. Religious adherents must follow the following precepts to "modernize" their religious consciousness: (a) they must inhabit the critical and self-correcting approach of science and accept "better arguments", (b) acknowledge that they have one faith among many, and (c) accept functional differentiation that threat all religions equally. Religions that meet the challenges of modernity must be consolidated, and those that do not must be reflexively neutralized (i.e., adapted in dialogue with contemporary criticism). Religion that aligns itself with "archaic" values and ignorant fundamentalists must be confronted and critiqued (Skirbekk 2009, p. 87). Discourses from certain disciplinary fields (e.g., science) and exemplary social practices (critical reflection and argumentation) are therefore recontextualized to religious education to instigate reform in religions. By critically discussing the experiences of different peoples, conflicts can be reduced in school, according to Skirbekk. In this respect, Skirbekk agrees with Habermas (cf. Skirbekk 2009, p. 101) that religion should be taken seriously. Religion might contribute positively by offering discourses about existential and spiritual themes through "saving translations". Saving translations might be normative resources, feelings of community, and moral truths found in religions. Thus, Skirbekk is also somewhat empathic in his framing of the discourse, as a critique of religion also seems to be connected to human flourishing.

Skirbekk underlines the importance of the teachers' professional assessment in the teaching of the critique of religion. He nevertheless gives some preliminary pedagogical suggestions: (1) The teacher should teach about monotheistic religions in parallel, and emphasize common problems (such as the problem of evil), (2) the teacher should draw

actively from various perspectives from different disciplines and religions, (3) the teacher should avoid lecturing about details and focus on central aspects, like differences between religions (Skirbekk 2011, pp. 6–7). The goal is to develop a Kantian "reflexive" thought process and not a negative evaluation of religion. A critique based on comparative perspectives and deliberative dialogue is fronted in Skirbekk's discourse.

The second actor in the critical discourse, Brekke (2018, 2020) shares many of Skirbekk's tenets, and posits that critique of religion should center around (the Kantian notion) of self-enlightenment where the teacher supplies pupils with appropriate tools to escape their "self-imposed immaturity". By "self-enlightenment" people can emancipate themselves rather than be supplied with "correct answers". Accordingly, "irrefutable absolutes" should always be questioned. Brekke highlights that critique should not center on what is worthy of critique in a certain religion, but rather take the ideal model of a community of inquiry that seeks to understand a theme better as a point of departure (Brekke 2018, pp. 126–27). This strand in the discourse was advocated for by Winsnes almost 40 years prior—but there are no direct references between the two authors.

Brekke nevertheless appreciates Skirbekk's efforts to engage in co-current developments of religious and non-religious sentiments, but tones down the occasional polemical discourse (Brekke 2018, pp. 124–30). Specifically, Brekke wants to introduce hermeneutics of suspicion via Freud, Nietzsche, and Marx, which he proposes should "all give ample room for extending the individual pupil's analytical tools in meeting one's tradition and that of others" (Brekke 2020, pp. 268–69). Reflections about the critique of religion can have transferability to other parts of life, Brekke suggests (Brekke 2018, p. 127). A critique of religion is, in essence, a concurrent critique of human reason. This is because criticism of religion quickly touches upon questions and challenges that relate to humanity, such as being too certain of our positions, creating outgroups, and rationalizing precepts for instrumental gain. Religion is thus conceived of as a hermeneutical construction that, like the "old Greeks", should be "brought into the conversation" with the pupils. The "resource-oriented" perspective defining religion as an important "reservoir" of symbolic knowledge has many similarities to central actors in pedagogics of religion. The interdiscursive reproduction of the discourse showcases that although no direct intertextual references are made to them, discourses are nevertheless interconnected in a relational web (cf. Jørgensen and Phillips 2002, p. 73).

Brekke highlights the need to seriously engage with the *content* of the critique discussed, and here he diverges from Andreassen (outside-perspective discourse), who Brekke thinks focuses too much on the meta-theoretical perspective (Brekke 2018, pp. 123–24). In fact, Brekke engages with Andreassen's reflections on the critique of religion on multiple occasions. He points to the fact that Andreassen's critique of religion shall not "contribute to the students' philosophical development, but must at the same time have a formative function" by putting the pupils' perspectives in relief and thus creating more perspectival awareness (Brekke 2018, p. 122). He also points to another dimension that might complicate the teaching: since the critique of religion in the outside-perspective is based on "methodological atheism" (bracketing ontological questions in the classroom) it lacks a coherent or obvious normative guideline that can guide the foundations for the critique (Brekke 2018, p. 122). The critical "outside perspective" does not secure equal treatment of religions, according to Brekke (Brekke 2018, p. 118). These negotiations showcase that the authors in the field build on and contest each other, trying to establish their discourse on the critique of religion as the correct or dominant view.

The last researcher in the critical discourse is exemplified by Henriksen (2005, 2012), who writes two articles in a journal directed toward teachers of religious education. He suggests that critique of religion is important for religious people and society writ large. Therefore, schools should, with reference to concrete and relevant examples, give students different avenues and tools to criticize religion. He thinks that this approach is appropriate, because it forces the teacher to stay nuanced, historical, and dynamic, showcasing the complexities of religion. Conversely, criticizing religion in broad terms may paint with too

broad a brush, according to Henriksen. Emphasizing specific traditions, he admits, may feel threatening to students. But this is not framed as an issue; on the converse, criticizing religion is seen as taking the students seriously. Based on the scholar of religion Cora Alexa Døving, Henriksen highlights three principles for a critique of religion: 1. Critique should have a recipient (person, institution, text), 2. Critique should be clear and factual, 3. Critique should be constructive and have a goal to improve something. Henriksen thinks that such critique might create strong responses from students, but nevertheless emphasizes the critique must endure, especially if religion perpetuates wrongdoings or discrimination. This discourse frames attentive and serious confrontation as an educative goal.

In sum, the critical discourse advocate for a critique of religion based on (1) deliberative discussion that seeks to consolidate religious practices and beliefs in light of contemporary developments, (2) A Kantian reflexive critique (Brekke and Skirbekk) and a "constructive" negative critique of religion (in Henriksen), (3) a critique of religion based on interdisciplinary perspectives (and not exclusively on religious studies, as seen in the outside-perspective discourse) and (4) comparative analyses.

Schjetne (2014) represents the multicultural discourse in a book chapter from an anthology discussing the ethical aspects of teacher practices. In contrast to the other discourses discussed so far, the chapter mostly discusses the potential *pitfalls* related to an unnuanced criticism of religion, with reference to theories of identity and power. Schjetne believes that religious education must accommodate minorities, especially those inhabiting conservative religious viewpoints. They are already in a compromised position by espousing religious norms and rules that might differ from dominant ones. Moreover, they have great exposure to secular society, while many secular people will have very limited and meaningful experiences with conservative religious life (cf. Spinner-Halev 2000, p. 26). A critique of religion directed at conservative viewpoints would therefore be unproductive and unconstructive, as conservative religious people's conception of "the good life" may be characterized by older generations' experiences and guidance from cultural and religious traditions. A challenge of such narratives may be considered especially violative. This contrasts with secular pupils, that would be interested in being critiqued as this will make them able to "live their life in a way they experience to be right" (Schjetne 2014, p. 164). It therefore seems like Schjetne does not disincentivize teachers from criticizing secular pupils' worldviews, as he considers such criticism wanted communication in the students' lives.

Schjetne criticizes both Skirbekk (critical discourse) and Andreassen (outside-perspective discourse) for not including power dynamics and dimensions of identity in their discussions. He believes they have too narrow views on the critique of religion and that they treat religious education as a university subject. Schjetne believes that we must approach conflict perspectives and anti-hegemonic sentiments more descriptively, and points out that teachers can explore why someone thinks as they do. The critique of religion in Schjetne's framework is therefore not about "pointing fingers". Schjetne shows this through covert interdiscursive references to phenomenology, as he thinks that we should bracket our preconceptions about knowledge that we at first glance would find difficult or worthy of criticism. Pupils can, in fact, learn something from that which at first glance seems contradictory compared to established truths (Schjetne 2014, pp. 167–69). Teachers can establish a space for such "bracketing" in the classroom by finding the latest and most up-to-date theological justifications for "controversial" and non-dominant attitudes. Empirically, the teacher may discuss how prominent certain perspectives are in selected religious traditions.

Schjetne proclaims that teachers must strike a balance between perpetuating important "modern" and "liberal" values, while simultaneously staying open to the value of religious and non-liberal traditions. He advocates for an open and comparative discussion in the classroom where the lines between "criticizer" and "criticized" are blurred. The "hunt" for specific "bad" religious practices (negative critique of religion) therefore subsidies, and attention is directed towards oppression more broadly. Students should stay open to the contestations of different perspectives and the widening of their horizons. Thus,

the teacher can achieve a "double socialization" by acknowledging the identities of some students while simultaneously expanding it for others (building on Gravem 2004, p. 398). The "critical element" in the pedagogy lies in the fact that all religions are represented as equal arbiters of truth (cf. interreligious criticism of religion, see over). Pupils are therefore faced with the fact that most religions, and religious people, think of their own beliefs as correct. Terms from multicultural theory are recontextualized (Chouliaraki and Fairclough 1999, p. 104) into discussions of the critique of religion in religious education by Schjetne, potentially lifting formerly undiscussed practices and considerations into the limelight, such as power dynamics and identity formation (cf. Skrede 2017, pp. 34–35). Schjetne's approach arguably reproduces an "empathic" discourse on religion, but in contrast to the former discourses, special credence is not given because of religious "contents". It is rather suggested based on reflections on power and identity in a religiously pluralized Norwegian context. This showcases that although the same "signs" in a "field" might intertwine (such as empathy/critique) the *contents* of these central signs might be transformed by and in historical processes.

The critical-realist discourse fronted by Søvik (2011, 2018) is inspired by critical-realist ontology and epistemology (see over), and the theologian Andrew Wright. Criticism of religion is important for Søvik, as religion can (and has been) used to frame horrible actions as morally sound (Søvik 2011, pp. 56–57). A critique of religion, in contrast to dialogue about religion, is not only about understanding the "other", but concerns defending one alternative and criticizing another. Criticism can relate to both the truthfulness of religious claims, and the practical consequences of believing in religious claims. Teachers should bring forth different arguments regarding central claims in all religions to keep education pluralistic. Examples can range from whether Moses got the ten commandments from God, if Jesus was resurrected, or if humans are purely "physical", Søvik advises. He also challenges teachers to help students critically engage with the understanding of their worldviews (Søvik 2018, p. 224). The teacher should not propose what religion or worldview to favor, nor frame the discussion in such a way that they can be "settled" for good. The classroom should always strive for "better explanations". This will lead to enthusiasm, development of intercultural competencies and critical thinking skills, according to Søvik. Serious and high-stake discussions give epistemological insights and interpersonal considerations about values. They can also incentivize meta-cultural competence and respect by preparing students to engage constructively in existential themes. Critique can foster modesty, because it can make students (and teachers) aware of all the aspects of life that we do not know *that* we do not know. Søvik proposes that a serious discussion may be much more transformative in creating understandings of the "other" than symbolic discourse about "respecting" differences. This might be an implicit dig at the multicultural (and similar) discourse, that we have seen are more skeptical towards such discussions.

Søvik explains that everyone wants to get their identities acknowledged. It is inherent to human biology to seek comfort and stability in identities and communities. Nevertheless, he goes on to say: "It is also important that rationality is a value that society can gather around. Other considerations are more important than defending obviously problematic opinions. It cannot be a goal with the education that no-one shall change opinions" (Søvik 2018, p. 230). This pushes back against the empathic discourses on religion (for instance seen in the multicultural discourse), and sides with the discourses advocating to challenge "problematic" religious beliefs through a deliberative discussion of negative criticism of religion. Much like the critical discourse, Søvik is therefore interested in comparative and deliberative discussions and serious debate, but with a very specific ontological and epistemological starting point.

The fifth discourse that I call the praxis discourse is based on a textbook by Repstad and Tallaksen (2014). Under the heading of "Critique of religion" (Repstad and Tallaksen 2014, pp. 139–41), the authors relate their discussion to a quote from a student: "The critique of critique-worthy conditions in religions should be discussed in the classroom". It

therefore seems like also Repstad and Tallaksen advocates for a scrutiny of a *negative critique* of religion. They stress that teachers must not stray away from dealing with "hard areas in religion and worldviews", and elaborate on the specifics behind the official phrasing that education in religious education should be "objective, critical and pluralistic" by specifying that "teachers should [at least] raise some critical questions towards religions and worldviews". Furthermore, they stress that pupils must also raise a critical perspective inward toward their own worldviews and traditions. Critical questions must be informed by thorough background-knowledge, to give the pupil's a space "to position [critique] from". In a "calm conversation", pupils might discuss and question each other. Teachers should encourage pupils who represent the religion under investigation to engage actively in the discussion.

Repstad and Tallaksen base their proceeding discussion on three different intertextual references: Bengt-Ove Andreassen (the outside-perspective discourse), and Levi Geir Eidhamar & Geir Winje (both contributing authors in the formerly discussed Sødal 2001). From Andreassen, they sketch three forms of criticism of religion that teachers should be aware of: normative (external), religious (internal), and critique from one religion to another (interreligious critique). Based on an "unpublished article" from Eidhamar, they discuss "interesting aspects" concerning the critique of religion, involving a relationship between "case" and "person", power relations, minority-majority dynamics, and the relationships between critique and blasphemy.

Repstad and Tallaksen raise four questions which can be discussed about the critique of religion, based on the reflections of the scholar of religion Geir Winje:

1. Is the way the criticizer describes the religion accurate? Can this be documented?
2. Is the "critique-worthy" conditions representative of the religion or worldview as such? Or are conditions criticized not prescribed by the religion or worldview?
3. Are the conditions criticized also present in other religions, or are they only relevant for the religion or worldview that is criticized?
4. Which criteria are the criticized religion or worldview criticized from? Are the criteria taken from the religion or worldview itself, or are they taken from another value system? (Repstad and Tallaksen 2014, pp. 140–41)

The praxis discourse is therefore heavily influenced by concrete intertextual references to other authors, possibly a consideration aimed at being useful and quick for the intended readers. It is however unclear how the authors themselves interpret these intertextual references and how they should be employed in specific contexts.

The hermeneutical discourse exemplified by an article written by Fuglseth (2018) is heavily influenced by the phenomenological tradition of Husserl and Gadamer. Critique can be "wide" and "narrow", according to Fuglseth. Narrowly understood, critique is a negative evaluation (negative critique of religion). Broadly understood, critique is conceptualized as "description" or "analysis". Such "criticism" does not necessarily seem to be "negative", just as a movie can get "positive" criticism in a review. To understand a narrow understanding of critique, pupils must first have a broad understanding. Contextualization and thorough pre-understandings are therefore important. Furseth evaluates that it is better for pupils to "misunderstand" than to "not understand" critique. If you are introduced to (narrow) critique without context, you do not have a "horizon" to place the criticism in, and therefore you do "not understand". This is futile. However, if pupils are introduced to a theme beforehand, they might misunderstand a critique, but at least this involves a re-calibration and change in relation to their previous horizon of understanding (Fuglseth 2018, p. 170). Context is therefore important to properly understand a narrow criticism of religion.

Fuglseth seems to conceptualize critique on two levels. Firstly, critique is deemed a "precondition for learning". Based on Kate Meyer-Drawe, Furseth proposes that all learning involves change. The change is the learning, and learning always involves "re-learning". He goes on to say that "nothing is more changing than judging criticism [ . . . ] in this theory it is posed that learning is always 'negative' and thus critique is always positively evaluated"

(Fuglseth 2018, p. 171). Secondly, critique is related to learning through hermeneutics in three ways. First, criticism can contribute to the "expansion of horizons" as teachers can show pupils that what they have in their immediate "horizon" might be connected to other contexts than first assumed. A historical-critical critique of holy texts may trigger such expansion. Second, the critique can also make pupils aware that their horizons of understanding influence how they understand the world. Lastly, the critique can make pupils look at a matter from a new angle, which "opens new horizons" and thus showcases that there are multiple avenues of knowing. Like many authors discussed so far, Fuglseth seems to be engaging in a hermeneutical critique of religion, but instead of referencing the potential decentering intrinsic to being exposed to new horizons of understanding or interpretations, Furseth outlines his call for criticism of religion through direct groundings in hermeneutic theory.

Fuglseth thinks that pupils must be trained in "concrete critique", for instance by "trying out attitudes", "giving reasons for opinions and attitudes rationally in school" or asking challenging questions (Fuglseth 2018, pp. 173–74). This is paramount as "to practice critique, is to practice understanding. If we shall understand well, we must meet critics and train in criticizing" (Fuglseth 2018, p. 176). Importantly, teachers cannot know a priori which methods or procedures will instigate a widening of pupil's horizons—this must be negotiated locally. Critique must nevertheless neither deem everything to be wrong (skepticism) nor suggest that only one thing is right (absolutism) (Fuglseth 2018, p. 176). Instead, Furseth suggests a "third way" based on Gravem (2004) (note the similarities to the multicultural discourse here): Teachers must acknowledge that *something* can be true. They must in consequence be rigorously reflective, not only critiquing religion and non-religion, but also engage in a critique of the critique and critique of meta-critique (Fuglseth 2018, p. 176).

*3.3. The Critique of Religion Elaborated: 2020–2022*

14 years after LK06, a new curriculum aimed at preparing students for modern life emerged. This also included a new curriculum for religious education. The critique of religion is still explicitly mentioned in the curriculum for religious education in upper secondary school, but is omitted from primary school. Pupils in religious education in primary school might still encounter a critique of religion through "critical thinking", "critical evaluation of sources" and engagement with "dissent and disagreement" (Kunnskapsdepartementet 2022c, p. 8). In upper secondary school, the formulation is now that pupils should "discuss different forms of critique of religions and worldviews" (Kunnskapsdepartementet 2022a, p. 5).

I have found limited discourses in the field of religious education research discussing the critique of religion in this period, but one example is found in an introductive book aimed at religious education-teachers in primary school, edited by the scholar of religion Kåre Fuglseth and professor in education, Thor-André Skrefsrud (Fuglseth and Skrefsrud 2021). In one of the chapters written by Fuglseth (2021), the book discusses the critique of religion explicitly but briefly. The author stresses the importance of critical perspectives in textbooks, even though critique can affect individual pupils' beliefs (Fuglseth 2021, p. 154). Textbooks should not be afraid to comment on or criticize certain notions "in light of today's understanding of reality". Teachers should give concrete examples of negative criticism of religion, for instance by commenting upon different notions of "Hell" or stories where God kills men to make a better human race (Fuglseth 2021, p. 154). These notions must be handled with care, especially if they discuss themes with competing understandings (interreligious criticism of religion), according to the authors.

Another example is reflected in an article entry written by the scholar of religion Bøe (2020), who proposes that discussions of an *internal* critique of religion can be a way to show diversity in the classroom. Drawing from her expertise on feminism in Islam, she showcases how teachers might draw on conflicts and controversies constructively. She elaborates:

> The heterogeneity of Muslim views and interpretations is reflected in internal debates on Islam. Although contentious, there is an established tradition for

internal religious critique within Islam. Such internal debates involve critical discussions and examinations of religious texts, as well as the legal thinking and reasoning in Islam (Bøe 2020, p. 4).

Feminist readings in Islam can provide "in-depth insights and understandings of how gender issues are debated to a great extent within the religion. Moreover, it adds information on the method of internal religious critique existing within Islam". Discussions of feminist critique in the classroom can nuance the stereotypical image of Islam as patriarchal and highlight diversity outside of heteronormative frameworks (Bøe 2020, p. 9). Through thorough contextualization and analysis of power, internal feminist critiques may nuance "grand narratives" about Islam often perpetuated in the religious education classroom, according to Bøe. Interestingly, we can thus spot a continuation of two discourses discussed so far: One where "internal" critique of religion is emphasized (present in the pedagogics of religion), but now the focus is inverted "outwards" to minority religions. On the other hand, Bøe is preoccupied with discussions of the power dynamics present in the classroom (as we also saw in Schjetne/the multicultural discourse). Old discourses are therefore reformulated in new creative ways, showcasing the non-obvious interdiscursive connections in the contemporary discourse on the critique of religion (Skrede 2017, pp. 51–53).

## 4. Concluding Discussion

The analysis has shown how the discourses on the critique of religion in religious education have been represented and negotiated throughout the last decades. It is clear that the critique of religion is an "important sign" in the field of religious education research, and that its contents are pulled in multiple directions at different historical time periods. From 1976–1997 three forms of discourses can be located: Internal, external, and hermeneutical critique. Critique is internal in the sense that Christianity is sought to be developed, either in the form of an accommodating discourse seeking to "help" students develop their religious sentiments, or as a "modernizing" discourse seeking to represent Christianity as a "thinking faith". Here, we can witness a dialectical interaction between different critiques and discourses, as an external critique from non-religious actors such as "Christianity's enemies" or other social systems instigate an internal re-calibration and critique of "dated" practices. Hermeneutics and diversity of interpretation are also discussed as a constructive way to critique religion, for instance by showcasing the internal tensions in the Bible. Little attention is nevertheless paid to critique in general. The restricted emphasis on the critique of religion might reveal the intimate connection between dispositive and discourse as the critique of religion appears to be close to invisible in the older curricula (M74, M87).

Through KRL and the reworked integrative religious education (Alberts 2012), the critique of religion becomes more present in the official documents between 1997–2006. The references are still mostly implied, which might indicate that the critique of religion was potentially incongruent with the subject's overall goal of cementing stable identities and creating respect for others. A struggle to expand the content of the discourse on the critique of religion is nevertheless present in the field of religious education research after 1997. This "new" critique especially concerns the potential de-centering that lies in diverging accounts of existential and religious themes. Such critique might be called integrative/interreligious, entailing that religious narratives, practices, and beliefs directly or indirectly contradict or challenge each other. Sødal (2001) discusses the pedagogical challenges related to interreligious critique present in both aesthetic and narrative dimensions of education. The only text analyzed after L97 that does not seem to emphasize interreligious critique is Skrunes (1999). Like his theologian peers, he upholds the importance of internal critique, but re-negotiates these criticisms through his own readings of the bible. Stabell-Kulø (2005) also breaks with the order of the discourse by introducing a negative critique of religion. Sødal (2001) expand the boundaries for the discourse, as critique of religion is related to interpretations of genres, multi-voiced-ness of texts, and critical outside perspectives. This analysis contrasts with former descriptions of the discourses in this period of religious

education research, which has characterized the discourse after 1997 as mostly empathic (cf. Andreassen 2008a).

The relatively novel discourse about criticizing the "negative" role religion can play in society is largely re-negotiated in the discourse between 2006–2020. Although to different degrees of articulation, all authors suggest to actively engage with religion's role in conflicts, oppression, and perpetuation of values in discord with the Norwegian school. It seems that a new observable and semi-stable "order of discourse" which actively highlights the destructive aspects of religion through critique is sustained. What is especially interesting after LK06 is that actors who are not typically involved in publications on religious education and are not religious education researchers (e.g., Søvik, Henriksen, Skirbekk), suddenly are engaging in the debate. There is therefore a new polyphony of interdisciplinary voices in the "field" that suggests new modes of "doing" the critique of religion through "redesigning existing discursive practices" (cf. Fairclough 2010, p. 137). The discursive process of recontextualization is especially evident after LK06. Recontextualization is about the introduction of external discourses into new "fields", creating new "hybrid forms" in the discourse (Skrede 2017, p. 54). The discussions of multicultural theory (Schjetne), critical-realist epistemologies (Sødal), critical theory (Kant and Habermas), and religious-studies-based subject-ontologies (Andreassen) showcase how new important "signs" are extrapolated from new contexts and "mixed" with the discourse of religious education research, creating new normative groundings.

Furthermore, new genres appear alongside textbooks that formerly dominated the discourse before 2006. There are now also chapters in anthologies, peer-reviewed articles, and articles in journals read mostly by teachers (cf. Henriksen's article). Genres inevitably influence the ways discourse is structured, consumed, and (re)produced (Skrede 2017, pp. 34–35). The introduction of genres such as journal articles and teacher journals introduced new "ways of acting", potentially letting the actors in the field of religious education research pursue the critique of religion in more diverse ways (see Fairclough 2010, p. 75).

Interestingly, after LK06 there is also a much larger degree of direct opposition in the discourse. Brekke (critical discourse) disagrees with Andreassen (outside-perspective discourse) and Skirbekk (critical discourse, i.e., an "internal" negotiation). Schjetne (multicultural discourse) challenges Andreassen (outside-perspective discourse) and Skirbekk (critical discourse). Note that the authors without religious-studies backgrounds seem to be very critical of epistemologies or ontologies that are only based on religious studies (cf. Andreassen). Andreassen is conversely critical of those who want to withhold the "empathic" representations of religions, an import he traces to the historically confessional religious education. The discourse on the critique of religion can therefore showcase two important positions in the field of religious education research; one that wants to radically break with its theological roots and criticize religion through religious studies perspectives, and another which seeks to incorporate theology (and other disciplines) into a comparative and deliberative critique of religion. Other intertextual references in the discourse from 2006–2020 are not necessarily in opposition but rather (re)produce discourses from the other actors in the field by building on their publications: Repstad & Tallaksen (praxis discourse) reference Andreassen (outside-perspective discourse), and Søvik (critical realist discourse) mentions Schjetne (multicultural discourse) and Skirbekk (critical discourse) briefly. Søvik (critical realist discourse), Henriksen & Brekke (critical discourse), Furseth (hermeneutical discourse), and Repstad & Tallaksen (praxis discourse) seem to largely go unnoticed by the other authors, which may underline that their relational position in the "field" has made them excluded outliers in the discourse. The dynamics of exclusion do not seem to be based on disciplinary backgrounds (as some scholars who are included in the dominant discourse have religious studies backgrounds while some do not, some scholars in the critical discourse are included while some are excluded), but seem rather to be based on the specific histories of the actors within the field of religious education research (cf. Chouliaraki and Fairclough 1999, p. 101). The stratification of the discourse thus shows that the field was ready to introduce new voices into discussions about the cri-

tique of religion, but that many of these voices inhabit a position that is not (intertextually) recognized by the established voices in the discourse. We can thereby observe both stability and discontinuity in the discourse through dynamics of inclusion and exclusion. Herein also lies the potential for change, as outliers in the discourse such as Søvik propose novel ideas, such as discussing ontological matters through the critique of religion.

After LK20, the overt mention of the critique of religion is no longer included in the curriculum for primary school. This may be signalizing that the authors of the curriculum agree that some level of abstraction and maturity is required for constructive discourse on the theme. The critique of religion is nevertheless discussed in a textbook for primary school teachers, which reveals that the critique of religion is still deemed relevant for younger pupils. The maintenance of the critique of religion in the curriculum for upper secondary school also reflects the continuous relevance of criticism of religion in religious education. Bøe's contribution (2020) highlights one way that this relevance is translated into normative reflections. Bøe attempts to incentivize nuanced understandings of religion by highlighting power, representation, and oppression through an internal critique of religion, thus responding to Schjetne's (multicultural discourse) call to include power dynamics in the research of religious education.

In times of growing polarization and unnuanced criticism, society can gain from engaging in constructive forms of criticism of religion (Stenmark 2022, pp. 1, 11). Researchers of Norwegian religious education have thought deeply about how this can be done, as this article has displayed. The review article has systematized knowledge that for too long has been scattered, underread, and under-discussed. Although the scholarly reflections discussed here are mainly tied to religious education, they can also give important insight into how to criticize religion more broadly. Teachers who find the criticism of religion (or other controversial topics) challenging, might also constructively engage with the discourses discussed here to develop their professional practices by deepening their knowledge of the diachronic developments in the field of religious education research.

**Funding:** This research received no external funding.

**Conflicts of Interest:** The author declares no conflict of interest.

## Notes

[1]  Note that the book (Sødal 2001) is edited by Helje Kringlebotn Sødal, but the individual chapters are written by multiple authors (Ruth Danielsen, Levi Geir Eidhamar, Geir Skeie and Geir Winje). To avoid confusion, I will nevertheless refer to this book as one collective work (Sødal 2001) throughout this article.

[2]  In 2007, Norway was convicted in the Human Rights Court in Strasbourg for breaching Article two; "The right to Education". The verdict empasized the compromised position of minorities in a common subject were Christianity held such a strong position (see Lomsdalen 2019, for more on this).

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
