# Peer review of "50 Years of Criticizing Religion: A Historical Overview of Norwegian Religious Education"

_religions, doi:10.3390/rel13090781_

Round 1
Reviewer 1 Report
The article presents an interesting matter. However the way the author presents his/her argument shows several weak points.The issue of critique of religion is presented in a rather abstract and isolated review of relevant material in regard to critique, the article refers to a meta-discourses, does not give context in sufficient way. This makes it very difficult for readers not familiar with the Norwegian RE developments to get the author's points and to be able to make up his/her own mind about the judgments presented. Also the basis of the analyzed texts in cultural and pedagogical practice is lacking.
At least a short discussions of the materials (textbooks) should be presented in advance to make the article comprehensive for outsiders (cf. as example passage on Stablell-Kulø p 6).
The manuscript contains numerous language and spelling mistakes.
It is unclear for me to which discipline the text tries to make a contribution. If this would be RE, the article certainly would gain quality if the author at least in short would draw lines indicating consequences for the overall development of the disclipline.
Author Response
I would like to thank the reviewer for a constructive criticism of my review paper on the critique of religion in RE. I have tried to address the referee’s concern in the following way:
- I agree that the initial draft did not give sufficient information to non-Norwegian readers about the context. I have therefore tried to give more context where seen fit, for instance line 57ff:
“Norwegian RE is compulsory for all pupils in Norwegian primary school. For pupils who choose a general study program, RE is an obligatory subject in the third and last year of upper secondary school. The subject is geared both toward developing knowledge and shaping attitudes, and one of its main goals is to reduce the conflicts potentially produced by religion in a multicultural society (cf. Toft, 2019, 8)”.
- I have also given a bit more context about the materials analyzed to initiate the reader to a larger degree, for instance 369ff or 524ff.
- It is an important point that the contribution of the article should be made more clear, and this has been greatly elaborated in both the concluding discussion and the introduction (for instance 21ff):
The goal of this article is to contribute to the literature on RE by systematizing an area of research that is underdeveloped (cf. Schanke and Hammer, 2018, 182–183; Löfstedt, 2020, 5). This study is a summary of former research, but also builds the grounds for future research by highlighting diachronic developments in the field (cf. Krumsvik and Røkenes, 2016, 62). An important aspect of the summary is to bring research that has been unsystematized, scattered, and buried in journals to the attention of scholars. The review can also be useful for teachers or teacher educators looking for concrete ways to handle the theme of critique of religion in the classroom (or contentious issues more broadly), as teachers seem to struggle with teaching about the critique of religion (Andersland and Aukland, 2020; Löfstedt and Sjöborg, 2020, 143–145).
- The manuscript has been vigorously checked for spelling mistakes.
Reviewer 2 Report
1. The topic "criticism of religion" is fundamentally relevant. What should be elaborated and substantiated more strongly: To what extent can the international/European discourse in religious education on criticism of religion benefit from the Norwegian debate?
2. The methodological approach seems theoretical-hermeneutic: the presentation of the Norwegian debate is very close to the original texts. The significance of the discourse analysis for the sub-chapters and the overall conclusion is not made clear enough. The methodological procedure should be made clearer (e.g. with regard to central distinctions such as "From 1976-1997 three forms discourses can be located: Internal, external and hermeneutical critique." or "I have sorted the publications in this period (2006-2020) into six analytical "sub-discourses", connected and divided by their similarities, differences, arguments and backgrounds." - In part, this makes the distinctions appear impure and lacks selectivity: "Each analytical category is exemplified by one or more authors. This does not mean that there are clear boundaries between the discursive categories, or that the authors are sole representatives for the epistemological/ontological from which they position their analysis.") and used to structure the content-related contexts more strongly.
3. Overall, the strongly descriptive and rather additive analysis would have to be structured more strongly by relevant discourse categories in order to be able to work out more general findings/learning effects. Even in the chapter "Concluding Discussion", the generalisation of the results is not yet really successful. On the other hand, it could be shortened in some places where individual authors are quoted in detail, resulting in redundancies (e.g. the three authors in the larger category called the "Critical discourse").
4. Ambiguities in theoretical reference points (e.g. critical pedagogy and Marxism-Leninism) should be revised.

Author Response
I would like to thank the reviewer for a constructive criticism of my review paper on the critique of religion in RE. I have tried to address the referee’s concern in the following way:
- I agree that we the referee that it is necessary to show the relevance of the critique of religion beyond the Norwegian context. I have therefore tried to point to how this is an important discourse to look closer at, see for instance line 21ff which ties the article to more general contemporary challenges and international research:
The goal of this article is to contribute to the literature on RE by systematizing an area of research that is underdeveloped (cf. Schanke and Hammer, 2018, 182–183; Löfstedt, 2020, 5). This study is a summary of former research, but also builds the grounds for future research by highlighting diachronic developments in the field (cf. Krumsvik and Røkenes, 2016, 62). An important aspect of the summary is to bring research that has been unsystematized, scattered, and buried in journals to the attention of scholars. The review can also be useful for teachers or teacher educators looking for concrete ways to handle the theme of critique of religion in the classroom (or contentious issues more broadly), as teachers seem to struggle with teaching about the critique of religion (Andersland and Aukland, 2020; Löfstedt and Sjöborg, 2020, 143–145).
and further 34ff
Norwegian RE is a diverse case study which that showcases different positions and negotiations that has analytical value outside of the Norwegian national context. The case illustrates how researchers from different ontological and epistemological frameworks try to rectify a pressing issue in contemporary society; how to criticize re-ligion constructively. This is important because critique can effectively challenge power and change religious practices that perpetuate human suffering and oppression. It is also imperative to engage in nuanced criticism of religion, as unreasonable critique may reinforce stereotypical imageries. A balanced conversation about the role of religion is also essential. in post-secular contexts characterized by the increased visibility and contention around religion in both media discourses and the public sphere (cf. Löfstedt, 2020, 2). Recent international research has also emphasized a new demand for scholarly discourses to engage in constructive criticisms of religion (see Franck and Stenmark, 2019; Lövheim and Stenmark, 2020). The Swedish philosopher of religion, Michael Stenmark points to the importance of criticism of religion for liberal democracies: To have the right and possibility to criticize religions in public life is crucial for developing a healthy liberal democratic society… a liberal democratic society must allow people who adhere to different worldviews to criticize each other’s religious or non-religious beliefs, values, and practices but still maintain relations of har-mony and comity across diverse outlooks on human life and its ends within its boundaries (2022, 1). RE can lay an important foundation in this regard by teaching pupils how to criticize religion in constructive ways.
- I also concord with the view that the presentation is close the original texts. I have tried to rectify this descriptive approach by emphasizing the “content” of the discourses more strongly (see for instance Line 310ff, etc.). I have also emitted the quote: “Each analytical category is exemplified by one or more authors. This does not mean that there are clear boundaries between the discursive categories, or that the authors are sole representatives for the epistemological/ontological from which they position their analysis”. The intention with this quote was to clarify that the categories constructed should not be taken as representative for “Multiculturalism” as such (as Multicultural pedagogy is a diverse tradition). However, I understand that this might devalue the usefulness of the categories (making them too “ambiguous”), and I hope the new differentiation between them is now clearer (see line 524ff).
- I have tried to be more concise and consistent with the “content related” interdiscursive dynamics throughout the analysis. I hope that this will have the added effect of making the argumentation in the conclusion more robust. I have also tried to make it clearer where further argumentation and commentary is needed, such as at 590ff, 607ff, 186ff, etc.
- This concern has been addressed, and the paragraph has been elaborated on in the following way (cf. line 231ff);
Asheim and Mogstad also engage with external critical discourses that try to address the authoritarian and oppressive crosscurrents in religion through education. These external critiques are directed towards “religion” broadly and not necessarily Christianity specifically. Through discussions of “critical pedagogy” (a pedagogical framework from the 1920s with roots in the Frankfurt school), Asheim and Mogstad discuss how education could contribute to developing students’ critical view of contemporary society. They also draw on Marxist-Leninist perspectives, which are more concerned with challenging class structures through education. Lastly, through dialogue with German pedagogics of religion, they reflect on the emancipatory dimensions of education (Asheim and Mogstad, 1987, 13–14, 21, 163). The goal of these discourses is to critique ideology and reveal how education is influenced by the school’s structures as well as socioeconomic factors. Importance is also given to the liberation of the human mind from predefined authorities, and continuous struggles to change the status quo. The authors do not identify themselves with critical pedagogy and are skeptical of its central tenets (21–22, 163).
Reviewer 3 Report
This review essay is an important summary of almost fifty years of religious education (RE) in the Norwegian school system, specifically with regard to the inclusion of critiques of religion in the curricula. The author has clearly demonstrated the evolution of the inclusion of critique in the curriculum and has provided helpful analysis of the theoretical foundations and various "sub-discourses" of critique included in pedagogical reflection on critiques of religion and the goal of RE in Norway.
The author has set a clear scope for the topic of the review: fifty years of criticizing religion in Norwegian RE. The author's demarcation of specific eras or periods of this approach to critique helps to clarify larger trends in theological and religious education in conversation with broader academic and cultural trends. The author has engaged in substantial research of the relevant literature and has provided ample citations and references for readers to follow if they would like to engage in deeper research on their own.
The author does assume a level of familiarity with the Norwegian education system and debates about Norwegian RE that might not be present in readers outside of Norway. It would be helpful in the introduction to explain in a bit more detail what Norwegian RE is, why RE remains part of Norwegian primary and secondary education, why the critique of religion was added to RE in the first place, and why it continues to be a contested topic into the present.
In the article, the author refers to multiple theorists and commentators on RE in Norway, often without introducing them or explaining their significance or relevance to the theme of the article. In many cases, authors are referred to solely by their last name. At least for the primary figures in the article, it would be helpful to give their full name and also to give just a very brief reference to their position and/or relevance to the topic of the article.
In terms of the purpose of the article, besides offering an analysis of internal dynamics and debates within the Norwegian education system, what are the "larger stakes" of this discussion? What are some important connections to broader cultural, political, religious, and theological concerns and debates outside of Norway? In other words, why should readers outside Norway care about this topic, and what can they learn from these Norwegian debates and developments?
What follows are specific comments, questions, and suggestions for edits for specific parts of the article:
Line 373 – please give some context and explain the significance of Article 2 in Protocol I
Lines 391ff. – a more detailed definition of “Outside-Perspective Discourse” is needed here.
Lines 395ff. – a more detailed definition of “Multicultural Discourse” is needed here.
Lines 407ff. – a more detailed definition of “Critical-Realist Discourse” is needed here.
Lines 410ff. – a more detailed definition of “Praxis Discourse” is needed here.
Lines 413ff. – a more detailed definition of “Hermeneutical Discourse” is needed here.
Line 432 – it’s not clear what the author means by “religious differentiation” here.
Line 444 – please say more about Klafki’s reflection on Bildung and key problems, so readers will understand the reference and its relationship to this sub-discourse.
In the section on the six sub-discourses, it would be helpful to discuss each of these sub-discourses in the same order they were introduced in the numbered list (beginning on Line 391). For example, in the numbered list Multicultural Discourse is listed second and Critical-Realist Discourse is listed third, but in the ensuing discussion Critical-Realist Discourse comes second and Multicultural Discourse comes third.
Line 459 – please say more about the meaning of “saving translations.”
Lines 461ff. – the concept of the “reflexive” appears several times in this section, but it is never clearly defined, nor is its significance and relevance to the topic clearly explained.
Lines 588ff. – a brief summary of critical-realist ontology and epistemology would be helpful here for readers who aren’t already aware of these concepts.
Lines 608-609 – this might be a translation issue, but it’s unclear what the end of this sentence means, as it ends with “all the aspects of life that we do not know, we do not know.” Perhaps the author means “all the aspects of life that we do not know that we do not know.”
A more general note is to give the full title of a program or curriculum before using the abbreviation (e.g. KRL or LK06), as readers who do not speak or read Norwegian will not recognize or understand these abbreviations or acronyms.
Additionally, the author will be well-served by having the manuscript proofread for grammatical and typological mistakes, especially concerning punctuation for possessives, incorrect or missing prepositions, adjectives or nouns that should be adverbs, incorrect forms of words (e.g. "were" when "where" is meant or "threat" where "treat" is meant), incorrect pronouns (e.g. "that" where "who" or "whom" is meant"), general subject-verb agreement, etc.
Author Response
I would like to thank the reviewer for a very thorough reading. I also thank for the constructive criticism of my review paper on the critique of religion in RE. I have tried to address the referee’s concern in the following way:
- I agree that the review assumes a level of familiarity with the Norwegian debate, and it has been hard for me to gauge how much space to spend on this theme. I have nevertheless included a couple of sentences in the introduction to give readers a little more information on the matter. For instance, the following formulations have been added (see also smaller addittions throughout):
Norwegian RE is compulsory for all pupils in Norwegian primary school. For pupils who choose a general study program, RE is an obligatory subject in the third and last year of upper secondary school. The subject is geared both toward developing knowledge and shaping attitudes, and one of its main goals is to reduce the conflicts potentially produced by religion in a multicultural society (Toft, 2019, 8).
- I concur that information about the author’s names, disciplines and backgrounds in the field is paramount to this analysis, and the issue has been addressed (see elaborations on all introductions of the authors discussed).
- I also agree that the article needs a “grander context” that it can highlight, vis-à-vis international discourses and pressing contemporary challenges. I have tried to address this, for instance through the following additions (line 21ff)
The goal of this article is to contribute to the literature on RE by systematizing an area of research that is underdeveloped (cf. Schanke and Hammer, 2018, 182–183; Löfstedt, 2020, 5). This study is a summary of former research, but also builds the grounds for future research by highlighting diachronic developments in the field (cf. Krumsvik and Røkenes, 2016, 62). An important aspect of the summary is to bring research that has been unsystematized, scattered, and buried in journals to the attention of scholars. The review can also be useful for teachers or teacher educators looking for concrete ways to handle the theme of critique of religion in the classroom (or contentious issues more broadly), as teachers seem to struggle with teaching about the critique of religion (Andersland and Aukland, 2020; Löfstedt and Sjöborg, 2020, 143–145)… The case illustrates how researchers from different ontological and epistemological frameworks try to rectify a pressing issue in contemporary society; how to criticize re-ligion constructively. This is important because critique can effectively challenge power and change religious practices that perpetuate human suffering and oppression. It is also imperative to engage in nuanced criticism of religion, as unreasonable critique may reinforce stereotypical imageries. A balanced conversation about the role of religion is also essential. in post-secular contexts characterized by the increased visibility and contention around religion in both media discourses and the public sphere (cf. Löfstedt, 2020, 2). Recent international research has also emphasized a new demand for scholarly discourses to engage in constructive criticisms of religion (see Franck and Stenmark, 2019; Lövheim and Stenmark, 2020). The Swedish philosopher of religion, Michael Stenmark points to the importance of criticism of religion for liberal democracies: To have the right and possibility to criticize religions in public life is crucial for developing a healthy liberal democratic society… a liberal democratic society must allow people who adhere to different worldviews to criticize each other’s religious or non-religious beliefs, values, and practices but still maintain relations of har-mony and comity across diverse outlooks on human life and its ends within its boundaries (2022, 1).RE can lay an important foundation in this regard by teaching pupils how to criticize religion in constructive ways.
- The document has also been proof-read vigorously
Specific comments
- I have adressed the concern regarding Line 373 vis-à-vis article 2, the following footnote has been added; In 2007, Norway was convicted in the Human Rights Court in Strasbourg for breaching Article two; “The right to Education”. The verdict empasized the compromised position of minorities in a common subject were Christianity held such a strong position (see also Lomsdalen, 2019, for more on this).
- All the six “sub-discourses” are now presented in the same manner as in the “Numberered” list. I have also given each sub-discourse a more thorough description (line 523ff).
- “Religious differentiation” has been renamed "religious diversity"
- The following statement has been added regarding “key problems” “that is, pressing matters that must be addressed in a specific time and space, such as global warming, immigration, digitalization, etc"
- Saving translations has been elaborated in the following way: ” Saving translations might be normative resources, feelings of community, and moral truths found in religions”
- The statements about reflexivity have been clarified (reflexivity here refers to adaption “in dialogue with contemporary criticism”).
- A brief summary of critical-realist ontology and epistemology is now given in the “numbered” presentation of CR/Søvik.
- The sentence formulation has been rectified (Lines 608-609).
- The local abbreviations (e.g., KRL and LK20) is now also elaborated on the first time they are used.
Round 2
Reviewer 1 Report
The revision the author/s made are substantial, the text in this new version is much more comprehensive also to non Norwegian readers, the language has been improved likewise. Thus I fully support to publish this version in your journal.